# Dual-Modality Ultrasound Imaging of SPIONs Distribution via Combined Magnetomotive and Passive Cavitation Imaging

**DOI:** 10.3390/s25237171

**Published:** 2025-11-24

**Authors:** Christian Marinus Huber, Lars Hageroth, Nicole Dorsch, Johannes Ringel, Helmut Ermert, Martin Vossiek, Stefan J. Rupitsch, Ingrid Ullmann, Stefan Lyer

**Affiliations:** 1Department of Otorhinolaryngology, Head and Neck Surgery, Else Kröner-Fresenius Foundation, Section of Experimental Oncology and Nanomedicine (SEON), Professorship for AI-Controlled Nanomaterials (KINAM), Universitätsklinikum Erlangen, 91054 Erlangen, Germany; 2Institute of Microwaves and Photonics (LHFT), Friedrich-Alexander-Universität Erlangen-Nürnberg, 91058 Erlangen, Germany; 3Department of Microsystems Engineering, University of Freiburg, 79110 Freiburg im Breisgau, Germany

**Keywords:** ultrasound imaging, ultrasound phantom, superparamagnetic iron oxide nanoparticles, magnetomotive ultrasound, passive cavitation mapping, magnetic drug targeting, nanomedicine, acoustic cavitation, focused ultrasound

## Abstract

Superparamagnetic iron oxide nanoparticles (SPIONs) have shown promise across a wide range of biomedical applications, including targeted drug delivery, magnetic hyperthermia, magnetic resonance imaging, and regenerative medicine. In the context of local tumor therapy (Magnetic Drug Targeting, MDT) SPIONs can be functionalized with chemotherapeutic agents and accumulated at tumor sites using an externally applied magnetic field. To achieve effective drug accumulation and therapeutic efficacy, precise positioning of the accumulation magnet relative to the tumor is essential. To address this need, we propose a dual-modality ultrasound imaging approach combining magnetomotive ultrasound (MMUS) and passive cavitation mapping (PCM). MMUS detects magnetically induced displacements to localize SPIONs embedded in elastic tissue, while PCM monitors cavitation emissions from circulating SPIONs under focused ultrasound exposure. In addition to detection, PCM has the potential to enable feedback-based control of cavitation exposure, allowing cavitation parameters to be kept within a safe regime. The dual imaging modality approach was validated using standard phantoms and a complex carotid bifurcation tumor flow phantom fabricated via 3D printing. Experimental results demonstrate the first coordinated spatiotemporal imaging of MMUS and PCM within the same anatomical model, resolving the key bottleneck of SPIONs monitoring in blood vessels/tissue. This demonstrates the strong potential of complementary MMUS and PCM imaging for monitoring in preclinical and clinical MDT settings.

## 1. Introduction

Cancer remains one of the leading causes of mortality worldwide, second only to cardiovascular diseases [1,2]. Conventional treatment modalities include surgical resection, chemotherapy, and radiotherapy, often used in combination. However, these approaches are frequently associated with systemic toxicity, non-specific targeting, and adverse side effects. In recent years, there has been a growing emphasis on localized cancer therapies that aim to enhance therapeutic efficacy while minimizing damage to healthy tissue [3,4,5,6,7]. A key innovation in this domain is the use of micro- and nanoscale drug delivery systems. Advances in nanotechnology have significantly expanded the possibilities for cancer treatment, enabling the development of nanoscale drug carriers composed of liposomes, lipid-based formulations, synthetic and natural polymers, and inorganic nanoparticles. These nanocarriers offer a range of advantages due to their small size and surface tunability, including improved biodistribution, enhanced stability, prolonged circulation time, and controlled or stimuli-responsive drug release.

A major advantage of nanomaterials in oncology lies in their ability to leverage the Enhanced Permeability and Retention (EPR) effect, a phenomenon whereby nanoparticles passively accumulate in tumor tissue due to the presence of leaky vasculature and impaired lymphatic drainage [8,9]. This passive targeting mechanism facilitates increased local drug concentration within tumors, improving therapeutic efficacy while reducing systemic exposure. However, despite the promise of the EPR effect, the clinical translation of nanoparticle-based therapies remains limited. Challenges include variability in patient response due to tumor heterogeneity, concerns over long-term toxicity and biodegradability, and the complexity of large-scale, reproducible manufacturing processes that meet regulatory standards [10,11].

In addition to their therapeutic applications, nanomaterials have significant potential in diagnostics. They can function as contrast agents in imaging modalities (e.g., magnetic resonance imaging (MRI) [12], ultrasound [13]), fluorescent markers in optical imaging, or even exploit their inherent magnetic, optical, or acoustic properties for real-time detection and localization of disease [14,15,16]. The integration of therapeutic and diagnostic (theranostic) functions within a single nanosystem represents a powerful tool for personalized medicine. Theranostic nanoparticles enable simultaneous treatment and monitoring, allowing clinicians to adjust therapeutic strategies in real time based on individual patient response, thereby improving treatment outcomes and minimizing side effects.

Superparamagnetic iron oxide nanoparticles (SPIONs) offer distinct advantages over non-magnetic nanoparticles due to their magnetic manipulability [17,18,19]. This property enables their use in a variety of therapeutic applications, including Magnetic Drug Targeting (MDT) [20], magnetic hyperthermia (MH) [21,22], and magnetically triggered localized drug release [23]. When combining nanomaterials with sonosensitive characteristics (ability to generate cavitation in response to ultrasound), the particles can be utilized as a theranostic system [24,25]. In previous research, we demonstrated that our lauric acid-coated SPIONs (LA-SPIONs) exhibit sonosensitivity [26,27]. In the context of MDT, SPIONs are typically functionalized with chemotherapeutic agents and administered via various routes (intratumoral, intravenous, or intra-arterial). An external magnetic field is then applied at the tumor site to accumulate the SPIONs [20,28], thereby enhancing local drug concentration and therapeutic efficacy. To achieve effective drug accumulation and complete tumor perfusion, the external magnetic field has to be dynamically repositioned. To address this need, a precise SPION distribution monitoring system has to be used.

SPIONs are commonly imaged using MRI [12] or magnetic particle imaging (MPI) [29]. While MRI offers high-resolution soft tissue contrast, it is not suitable for real-time monitoring during MDT procedures due to interference from the external magnetic field and the potential for equipment damage. MPI, in contrast, provides direct, high-sensitivity imaging of SPIONs and is compatible with magnetic field manipulation. However, MPI remains in the experimental stage and is not yet widely available for clinical use. In contrast, ultrasound-based imaging techniques represent a clinically established, cost-effective, and widely accessible alternative [13]. Ultrasound does not interfere with magnetic targeting systems and is already integrated into many clinical workflows. Moreover, by exploiting the acoustic responsiveness of SPIONs, including their ability to induce cavitation under focused ultrasound (FUS) exposure [26,27], ultrasound offers a promising platform for non-invasive monitoring of SPION distribution and therapeutic activity during MDT. Additionally, magnetic nanoparticles have been extensively investigated in combination with ultrasound for a wide range of biomedical applications, including ferrogels ultrasonography, acoustic characterization of tissue-mimicking phantoms, and ultrasonic transducer development [30,31,32].

Magnetomotive ultrasound (MMUS) [33,34] has demonstrated its utility in monitoring SPION accumulation during MDT procedures [35]. A key advantage of MMUS is its compatibility with existing MDT setups, requiring only an additional ultrasound transducer and no specialized imaging equipment. MMUS operates on the principle that tissue embedded with SPIONs undergoes mechanical displacement when exposed to an external magnetic field. By applying an alternating magnetic field, the resulting periodic tissue displacements can be detected using ultrasound to map the spatial distribution of SPIONs. In simplified scenarios, the MMUS displacement can also be used to estimate the SPION concentration [36]. However, MMUS is limited to imaging SPIONs embedded in tissue. This means it cannot detect SPIONs circulating in blood vessels, as the absence of mechanical coupling prevents measurable displacement, which will also be evaluated in this contribution. To address this, the sonosensitivity of our LA-SPIONs can be exploited for therapy monitoring via passive cavitation mapping offering a complementary imaging strategy alongside MMUS.

As previously described, sonosensitive SPIONs refers to the phenomenon whereby certain SPIONs can induce cavitation when exposed to a FUS field. Acoustic cavitation involves the formation and dynamic behavior of gas bubbles in a liquid medium in response to acoustic pressure fluctuations [37]. This occurs when the negative pressure of an ultrasound wave falls below the vapor pressure of the medium, or when localized heating causes temperatures to rise above the boiling point. The process includes the nucleation of new gas bubbles and the expansion of existing gas pockets to macroscopic sizes. Once nucleated, these bubbles exhibit nonlinear oscillatory dynamics that can be categorized as either stable or inertial. In stable cavitation, bubbles oscillate around an equilibrium size without collapsing, whereas in inertial (or transient) cavitation, the bubble undergoes rapid expansion beyond a critical radius, followed by a violent collapse. These cavitation events generate significant thermal, biochemical, and mechanical effects, which are of particular interest in biomedical applications [38,39,40,41,42]. In the context of localized chemotherapy, stable cavitation enhances vascular permeability and promotes the extravasation of drug-loaded nanoparticles from the bloodstream into the tumor. Inertial cavitation, on the other hand, can directly damage tumor cells or trigger the release of chemotherapeutic agents from SPIONs at the tumor site, enabling targeted and controlled therapy [42,43]. Beyond their therapeutic effects, cavitation events also emit characteristic acoustic signals. Inertial cavitation produces broadband noise, while stable cavitation generates harmonics, subharmonics, and ultraharmonics of the driving ultrasound frequency. These acoustic emissions can be captured and spatially resolved using active [44,45] or passive [46,47,48,49] cavitation mapping techniques (ACM and PCM, respectively). As such, cavitation mapping enables indirect but effective monitoring of sonosensitive SPION distribution in vascular environment, supporting their use in real-time, image-guided theranostic systems.

In this study, we assess the combined use of PCM and MMUS for comprehensive monitoring of SPION distribution in biomedical applications. To facilitate this, we utilize standard tissue and flow phantoms and introduce a sophisticated tumor flow ultrasound phantom capable of simulating various mechanical parameters, flow dynamics, and tumor geometries. Additionally, we employ a novel MMUS technique based on global time delay estimation [50] to enhance imaging accuracy and provide more detailed insights into SPION localization.

## 2. Materials and Methods

### 2.1. Superparamagnetic Iron Oxide Nanoparticles

The nanoparticles used in this work are lauric-acid-coated superparamagnetic iron oxide nanoparticles (LA-SPIONs), synthesized according to the protocol from [20,51]. This formulation has been extensively characterized in earlier studies, and the reported physicochemical and magnetic properties directly apply to the material used here.

Transmission electron microscopy measurements in [51] show that the magnetite cores have a diameter of approximately 7–8 nm, forming multicore aggregates embedded in a lauric-acid matrix. Dynamic light scattering measurements of the same formulation report a hydrodynamic diameter on the order of 100 nm with low polydispersity. Magnetic characterization using vibrating sample magnetometry and superconducting quantum interference device susceptometry demonstrates superparamagnetic behavior at room temperature with negligible hysteresis and a blocking temperature below physiological conditions [20]. The saturation magnetization is reported to be approximately 450 kA/m for this formulation, and the absence of remanent magnetization outside an applied magnetic field is confirmed, which is essential for biomedical applications.

### 2.2. Ultrasound Phantom Fabrication

Ultrasound phantoms are commonly used to replicate the tissue properties, geometry, or flow structures of real biological environments. These phantoms are typically constructed from tissue-mimicking materials (TMMs), such as agar, gelatin, polyvinyl alcohol (PVA) cryogel [52]. For MMUS imaging different types of ultrasound phantoms are utilized, but most are based on either a gelatin mixture [53,54,55] or on PVA cryogel [36,56,57,58]. Others, such as agarose possess gel structures with pores too large to retain the SPIONs, leading to poor spatial confinement and diffusion. This would lead to no trackable tissue displacement within the phantom.

However, since this study also employs PCM in combination with MMUS to image the distribution of SPIONs, phantom design must be adapted accordingly. Gelatin based phantoms can not be used for an extended period of time when submerged in water, which is the case of the PCM experiment. PVA based phantoms do not have this problem. Due to these reasons, PVA based ultrasound phantoms are utilized.

To evaluate the imaging capabilities of both MMUS and PCM, particularly under flow conditions, we utilize three different phantom types: a tissue phantom, a flow phantom and one complex model (see Figure 1). The first standard phantom consists of a homogeneous tissue-mimicking matrix with embedded SPIONs to assess static distribution. The second standard phantom is a simple flow phantom designed to accommodate various fluid types for dynamic imaging assessment. The third, complex phantom integrates both tissue-mimicking properties and flow channels, simulating a more anatomically realistic environment. The complex phantom represents a carotid bifurcation with a tumor.

The preparation of our ultrasound phantoms based on PVA cryogel follows a well-established freeze–thaw cycling process [58,59]. The phantom composition consists of 10 wt% PVA powder (Kuraray, Elvanol 71-30, Chiyoda, Japan), 1 wt% silica gel microparticles (Sigma Aldrich, St. Louis, MO, USA; serving as acoustic scatterers), and 89 wt% ultrapure water. Initially, the PVA powder is gradually dissolved in water by heating the mixture to 80 °C under continuous stirring until a clear, homogeneous solution is obtained. Once fully dissolved and transparent, the silica microparticles are added and mixed thoroughly to ensure uniform dispersion throughout the solution.

The resulting mixture is then poured into molds and subjected to a controlled freeze–thaw cycle (FTC), wherein it is frozen at approximately −4 °C for 12 h and subsequently thawed at room temperature (approx. 20 °C) for another 12 h. Each complete freeze–thaw cycle enhances the gel’s mechanical stability and mimics soft tissue elasticity by promoting crystallite formation within the polymer matrix. The number of FTCs can be varied depending on the desired stiffness and acoustic properties of the phantom [60]. In this study, two freeze–thaw cycles were applied for all phantoms to achieve suitable tissue-mimicking characteristics for MMUS and PCM imaging.

#### 2.2.1. Standard Tissue Phantom

The standard phantom employed for tissue bound SPION assessment consists of a rectangular outer block measuring 60 × 40 × 25 mm, with a central cylindrical cavity (10 mm diameter, 20 mm height) serving as the inner region (see Figure 1a). The outer region is composed of ultrapure water with 10 wt% PVA cryogel and subjected to two FTCs. In contrast, the inner region contains a SPIONs suspension in place of ultrapure water. The SPIONs were synthesized in-house at the Section for Experimental Oncology and Nanomedicine (SEON) and consisted of LA-SPIONs, which are suitable for MDT applications.

A stock suspension with a concentration of 12.85 mg Fe/mL was diluted with ultrapure water to prepare five test concentrations: 0, 0.1, 0.5, 1.0, and 5.0 mg Fe/mL. These concentrations were embedded into the inner cylindrical region of the phantom. Fabrication involved preparing the inner and outer components separately in custom molds, each undergoing one freeze–thaw cycle. Following this, the inner cylinder was inserted into the outer block, and the assembled phantom was subjected to a second FTC to promote mechanical integration between the two regions. For each iron concentration, two identical phantoms were produced, resulting in a total of ten phantoms.

#### 2.2.2. Standard Flow Phantom

The standard phantom used for flow measurements is also fabricated from 10 wt% PVA cryogel to ensure similar properties as the tissue phantom. In addition to simulating soft tissue, it incorporates a cylindrical flow channel to enable dynamic fluid experiments. To create the channel, a 2 mm diameter rod is positioned within the mold during the gel casting process. Following completion of two FTCs, the rod is carefully removed, leaving behind a well-defined flow channel.

The final phantom dimensions are 60 × 40 × 65 mm, with a 2 mm diameter cylindrical channel running along the depth axis (see Figure 1b). The position of the 2 mm flow channel is directly in the middle of the phantom. This configuration allows for controlled perfusion of fluids through the phantom, supporting evaluation of MMUS and PCM under flow conditions. SPION suspension with 0, 0.1, 0.5, 1.05, and 5.0 mg Fe/mL is used in the flow experiments.

#### 2.2.3. Complex Flow Tumor Phantom

The fabrication of the complex flow-tumor phantom involves a multi-step process adapted from our previous work [61,62,63], with modifications to accommodate PVA cryogel and the omission of the tumor dissolution step. First, the vascular bifurcation structure is 3D printed using a X1 Carbon printer (BambuLab, Shenzhen, China) with high-impact polystyrene (HIPS) filament (Keycoon GmbH, Nunus Filamente, Frankfurt am Main, Germany), which serves as a dissolvable scaffold (see Figure 1c for geometric details). Simultaneously, a two-part mold for the ellipsoid tumor geometry is printed in acrylonitrile butadiene styrene (ABS; BambuLab). This mold is designed to be securely attached to the flow bifurcation scaffold and removed after the first freeze phase.

To form the tumor region, a suspension of 1 mg Fe/mL SPIONs is mixed with 10 wt% PVA and poured into the ABS tumor mold fixed along the HIPS flow structure. This assembly is then frozen for 12 h to initiate the first FTC. After freezing, the tumor mold is carefully removed, and the entire structure is embedded in room temperature water–PVA solution to prevent premature thawing of the SPION-PVA tumor region. This approach ensures that both the tumor and surrounding tissue regions undergo identical FTC, resulting in consistent mechanical and acoustic properties across the phantom [58]. The composite structure is then frozen for 12 h to complete the first FTC. To finalize the two FTC, a second FTC is undergone.

The flow characteristics in this study were validated using ultrasound for imaging purposes [61,62,63]. For quantitative hemodynamic benchmarking (e.g., Reynolds number or shear stress), synchrotron X-ray PIV [64,65] is an appropriate complementary technique and will be considered in future work due to its ability to resolve multiphase velocity and density fields with high accuracy.

### 2.3. Magnetomotive Ultrasound

Magnetomotive ultrasound relies on tracking displacement of tissue infused with SPIONs, which react to the applied magnetic force(1)$$\begin{array}{c} F{\left(r,t\right)}_{mag\text{-}motive}=\frac{\chi_{S}V}{2\mu_{0}}\nabla \left({\left|B(r,t)\right|}^{2}\right), \end{array}$$
with the magnetic flux vector field $B={\left(B_{x},B_{y},B_{z}\right)}^{T}$ at position $r={\left(x,y,z\right)}^{T}$ and time *t*. $\chi_{S}$ is the magnetic susceptibility, $\mu_{0}$ the permeability constant, and *V* is the volume of magnetic core in each SPION that is influenced by the magnetic field. To map the distribution of SPIONs using MMUS, different approaches can be utilized, which all try to find the induced pattern of magnetic force within the recorded data.

Early MMUS techniques utilized various types of magnetic field excitation, broadly classified as either pulsed [66] or periodic [33]. Pulsed MMUS resembles elastography, requiring minimal data by comparing tissue displacements with and without magnetic field application, where different displacement estimation methods can be used [67]. In contrast, the standard approach employs a periodic sinusoidal magnetic field, which induces a characteristic oscillatory displacement at a known frequency [33]. This can be further optimized using coded excitations [68,69], such as chirp or Barker sequences, to enhance signal detectability. The most widely adopted method is the frequency- and phase-sensitive MMUS [57]. In this approach, SPION-induced displacements are first isolated based on their frequency content, and then differentiated from background motion through their distinct phase signature. SPION-driven motion exhibits a phase shift of approximately π radians relative to unrelated motion, providing a reliable basis for separation and mapping.

In this study, we implement a combined approach that integrates global ultrasound elastography (GLUE) [50] with frequency- and phase-sensitive MMUS [57], referred to as the frequency- and phase-sensitive global MMUS method. This technique enables subsample displacement tracking in both axial and lateral directions. Beamformed radiofrequency (RF) data are processed frame-by-frame, and inter-frame displacements are estimated using the GLUE framework. The method considers all samples across axial positions $a_{i}$ and lateral positions $l_{i}$ for i=1,…,M samples and j=1,…,N RF lines, using two RF datasets ($s_{1}$ and $s_{2}$). An initial axial A(i,j) and lateral L(i,j) displacement map is computed using a 2D dynamic programming approach [70], which is then further refined by minimizing this cost function:(2)$$\begin{array}{c} C_{S}\left(\Delta a_{1,1},\ldots ,\Delta a_{m,n},\Delta l_{1,1},\ldots ,\Delta l_{m,n}\right)\\ =\sum_{j=1}^{n}\sum_{i=1}^{m}\left\{{\left[s_{1}\left(i,j\right)-s_{2}\left(i+a_{i,j}+\Delta a_{i,j},j+l_{i,j}+\Delta l_{i,j}\right)\right]}^{2}\right.\\ \;\;\;\;\;\;\;\;\;\;\;\;\;\;\;+\alpha_{1}{\left(a_{i,j}+\Delta a_{i,j}-a_{i-1,j}-\Delta a_{i-1,j}\right)}^{2}\\ \;\;\;\;\;\;\;\;\;\;\;\;\;\;\;+\beta_{1}{\left(l_{i,j}+\Delta l_{i,j}-l_{i-1,j}-\Delta l_{i-1,j}\right)}^{2}\\ \;\;\;\;\;\;\;\;\;\;\;\;\;\;\;+\alpha_{2}{\left(a_{i,j}+\Delta a_{i,j}-a_{i,j-1}-\Delta a_{i,j-1}\right)}^{2}\\ \;\;\;\;\;\;\;\;\;\;\;\;\;\;\;+\beta_{2}{\left(l_{i,j}+\Delta l_{i,j}-l_{i,j-1}-\Delta l_{i,j-1}\right)}^{2}\}, \end{array}$$
where $\Delta a_{i,j}$ and $\Delta l_{i,j}$ represent sub-sample displacement corrections. The regularization parameters $\alpha_{1}$ and $\alpha_{2}$ enforce smoothness constraints on axial displacement along the axial and lateral directions, while $\beta_{1}$ and $\beta_{2}$ do so for lateral displacement. In this study, only the resulting axial displacement is utilized, with $\alpha_{1}=1$ and $\alpha_{2}=0.01$.

The cost function is linearized via a Taylor series expansion, resulting in a linear system of equations that is solved using numerical optimization. For this purpose, we employed an open-source implementation available at https://users.encs.concordia.ca/~hrivaz/Ultrasound_Elastography/ (accessed on 3 September 2025). This method enables precise axial displacement estimation between two frames (RF datasets). The process is repeated across all frames k=1,…,K, with each frame compared to the first, yielding 2D displacement maps over *k* frames A(i,j,k).

For each pixel (i,j) a *N*-point discrete Fourier transform (DFT) along *k* for a single frequency $f_{0}$ computes(3)$$\begin{array}{c} U\left(i,j,\ell_{0}\right)=\sum_{k=0}^{K-1}A\left(i,j,n\right)e^{-j2\pi n\ell_{0}/N}, \end{array}$$
where $\ell_{0}=\mathrm{round}\left\{f_{0}/\left(f_{r}\cdot K\right)\right\}+1$ is the frequency-domain index and $f_{r}$ is the frame rate. $U_{R}\left(i,j\right)$ and $U_{I}\left(i,j\right)$ are the real and imaginary part of the DFT result $U(i,j,\ell_{0})$. Utilizing the real and imaginary part, the peak to peak displacement (see Figure 2a) is(4)$$\begin{array}{c} u\left(i,j\right)=4\sqrt{{\left(\frac{U_{R}\left(i,j\right)}{K}\right)}^{2}+{\left(\frac{U_{I}\left(i,j\right)}{K}\right)}^{2}}. \end{array}$$The corresponding phase (see Figure 2b) is the argument of the complex DFT result:(5)$$\begin{array}{c} \phi \left(i,j\right)=\arg\left\{U_{R}\left(i,j\right)+jU_{I}\left(i,j\right)\right\}. \end{array}$$The phase ϕ is then used as mask for the final displacement map $u_{\phi }\left(i,j\right)$ (see Figure 2c), where only amplitudes corresponding to the same phase as the applied magnetic field are shown. The whole process is depicted in Figure 2. The phase of the magnetic field is subtracted from the phase computed in Equation (Equation 5). Only displacements corresponding to a phase difference below π/8 are utilized.

The utilized measurement setup is demonstrated schematically in Figure 3. The ultrasound transducer (L11-5v Verasonics, $f_{c}=7.6$ MHz center frequency, 128 elements, pitch 0.3 mm and 77% relative bandwidth) is connected to a research ultrasound system (Verasonics Vantage 64LE, $f_{s}=31.25$ MHz sampling frequency and $f_{r}=100$ Hz repetition frequency) and fixed on a linear positioning unit to scan the ultrasound phantom in 3D along the *y*-axis. Below the ultrasound transducer, the corresponding ultrasound phantoms are placed and acoustically coupled with ultrasound gel. Below the phantom the custom made MDT electromagnet (Siemens AG) is placed [20,71] (see Appendix A for further information), which induces a magnetic field with a sinusoidal time function with an offset of half the amplitude to only have positive values. The frequency of the applied magnetic field is $f_{0}=1$ Hz and has a maximum magnetic flux density of approx. $\hat{B}=1$ T directly on the pole tip and approx. 0.4 T at z=20 mm. The respective magnetic flux density gradient $\partial B_{z}/\partial z$ is around 90 T/m and 15 T/m. For each frame, 10 s of data with a repetition frequency of $f_{r}=100$ Hz are acquired. The linear positioning system displaces the transducer in 0.5 mm steps along the flow direction in the *y*-direction from y=0 mm to y=50 mm. It starts slightly before the tumor to still incorporate only flow structures and ends slightly after the tumor (see scanning scheme in Figure 1d). The 3D scanning scheme is only performed for the complex flow phantom. For the standard phantoms, only one frame of data is acquired and processed.

### 2.4. Passive Cavitation Mapping

To passively reconstruct the cavitation emission generated from SPIONs under the influence of a FUS field, the received radio frequency signals $s_{n}\left(t\right)$ are time delayed with $\tau \left(x\right)=\sqrt{z^{2}+{\left(x_{n}-x\right)}^{2}}/c$ for n∈N<N at point x=(x,y,z) and beamformed, with, e.g., the delay and sum beamformer (DAS):(6)$$\begin{array}{c} q\left(x,t\right)=\frac{1}{N\alpha_{n}}\sum_{n=1}^{N}s_{n}\left(t+\tau \left(x\right)\right), \end{array}$$
with $\alpha_{n}$ as the piezoelectric coefficient characterizing each array element. In PCM, a cavitation map of source intensity is generated by integrating the square of the source strength over a time period *T* with uniform array weighting in a discretized manner:(7)$$\begin{array}{c} I\left(x\right)=\frac{1}{\rho c}\sum_{m=1}^{M}q{\left(x,m\cdot \Delta t\right)}^{2}. \end{array}$$Equations (Equation 6) and (Equation 7) present the standard time exposure acoustic PCM (TEA-PCM) method. This method can be further improved using other beamformer such as the Robust Capon Beamformer [72], the Delay Multiply and Sum Beamformer (DMAS) [73] with higher orders [74], or the coherence factor [75]. Additionally, the method can also be implemented in the frequency domain [76], which reduces computational load and allows for the selective use of only the frequencies of interest.

The higher order DMAS is a correlation extension of the standard DAS beamformer. It incorporates higher order correlations between received signals from the transducer array, leading to improved imaging quality in certain scenarios. In this contribution, the third order DMAS (DMAS3) is utilized, which is expressed as(8)$$\begin{array}{c} q_{DMAS3}\left(x,t\right)=\frac{1}{N\alpha_{n}}\frac{e_{1}^{3}-3e_{1}e_{2}-2e_{3}}{6} \end{array}$$
with the power sums(9)$$\begin{array}{c} e_{j}=\sum_{n=1}^{N}{\hat{s}}_{n}^{j}\left(t-\tau \left(x\right)\right). \end{array}$$${\hat{s}}_{n}^{j}={\left(\mathrm{sign}\left\{s_{n}\right\}\cdot \sqrt[{j}]{|s_{n}|}\right)}^{j}$ is the unit corrected signal. Using the expression Equation (Equation 8) with power sums in Equation (Equation 9) allows for linear computational complexity of the DMAS3 computation.

The measurement setup used for PCM of cavitation activity induced on SPIONs is depicted in Figure 4. The entire system is submerged in a water tank to ensure proper acoustic coupling. The same linear array ultrasound transducer (L11-5v) and phantom models used in the MMUS experiments are employed. In this configuration, the MDT electromagnet is removed due to incompatibility with the water tank setup.

A custom-built FUS transducer is integrated into the system to induce cavitation. Acoustic absorbers are placed along the tank boundaries to suppress reflections and prevent standing wave formation. The FUS transducer has a center frequency of $f_{\mathrm{FUS}}=835$ kHz, driven by a 60 dB power amplifier (BONN Elektronik BTA 0122-1000, Holzkirchen, Germany) and a signal generator (Agilent 33220A, Santa Clara, CA, USA). A 3 dB attenuator is inserted between the amplifier and FUS transducer to optimize impedance matching.

The excitation signal consists of a single burst at $f_{\mathrm{FUS}}$ with a pulse duration of $\tau_{\mathrm{burst}}=40/f_{\mathrm{FUS}}$ = 47.90 μs. This yields a peak negative pressure of approximately 0.65 MPa, corresponding to a mechanical index (MI) of 0.7. The MI was deliberately kept below 0.7 to avoid inducing inertial cavitation in the absence of cavitation nuclei, consistent with established safety thresholds in the presence of gas bodies (e.g., contrast agents) [77,78].

The linear array transducer passively receives acoustic emissions generated by cavitation and transmits them to a Verasonics Vantage 64 LE system for data acquisition. Signal processing is carried out using the DMAS3-based PCM algorithm. For each location, 10 FUS pulses are delivered and recorded to enhance signal reliability.

In the case of standard phantoms, cavitation is measured at a single position, while the complex tumor flow phantom is scanned along the *y*-axis in 0.5 mm increments. A peristaltic pump (Pharmacia LKB P-1, flow rate of 5 mL/minute in a tube with a inner diameter of 2.1 mm) circulates SPION suspensions of varying iron concentrations through the flow channels. To quantify cavitation activity, the intensity of cavitation within the PCM maps is analyzed. Only the regions corresponding to the flow channels are considered for computing the mean and standard deviation of cavitation intensity.

## 3. Results

The fabricated phantoms serve different purposes depending on the imaging modality and evaluation purpose. Standard tissue and flow phantoms were employed to evaluate the behavior of tissue-bound SPIONs and SPION suspensions separately for MMUS and PCM imaging. The corresponding results are presented in Figure 5, showing the MMUS and PCM images for both phantom types using 1 mg Fe/mL of LA-SPIONs embedded in tissue or suspended in flow.

In the tissue phantom, frequency- and phase-sensitive global MMUS successfully visualized the SPION inclusion, producing a maximum displacement of approximately 1.2 μm. In contrast, for the flow phantom, the MMUS method did not detect displacement within the flow channel itself but rather just beneath it, with a reduced displacement amplitude of approximately 0.3 μm.

For PCM imaging, the results show an inverse trend compared to MMUS. When SPIONs are suspended within the flow channel, the applied FUS field successfully induces cavitation events, which are detectable through PCM. However, no cavitation was observed within the tissue phantom. This absence may result from the phantom preparation process or from insufficient fluid content within the tissue matrix, which is necessary for the nucleation of gas-filled bubbles. Since cavitation requires the presence of a liquid medium to form and sustain the bubbles, its absence limits the effectiveness of PCM in solid or semi-solid matrices.

Furthermore, PCM did not detect cavitation signals within the SPION-loaded region of the tissue phantom but occasionally outside of it. These signals are likely attributable to reflected echoes from residual air bubbles within the PVA phantom. However, the measured cavitation intensity in these regions was two orders of magnitude lower than intensities recorded during confirmed cavitation events. This significant difference allows for effective thresholding of the PCM data, ensuring that low-level signals within the tissue phantom are not misinterpreted as genuine cavitation activity.

In Figure 5, the relationships between MMUS displacement and iron content, as well as between cavitation activity and iron content, are illustrated. LA-SPIONs were tested at concentrations of 0, 0.1, 0.5, 1.0, and 5.0 mg Fe/mL within both flow channels and tissue inclusions. The MMUS displacement increases approximately proportionally with iron content within the tested concentration range, consistent with the magnetic force expression in Equation (Equation 1), where the force is directly proportional to the magnetic susceptibility $\chi_{S}$, which itself scales with iron concentration. Displacements measured in the tissue-bound phantom were approximately four times greater than those observed beneath the flow channel. It was possible to image tissue bound SPIONs to a iron content of 0.5 mg Fe/mL. At 0.1 mg Fe/mL, the displacement (estimated to be <0.1 μm) was not strong enough to be distinguished from the overall movement. This was the same for SPIONs in flow channel. This difference may be attributed to the total volume or distribution of SPIONs in the respective configurations.

In contrast, cavitation activity within the tissue phantom was negligible, indicating an absence of detectable cavitation events. In the flow phantoms, however, cavitation activity was sufficiently high for effective mapping for all iron contents, even down to 0.1 mg Fe/mL. The relationship between iron content and cavitation activity did not follow a linear trend but rather exhibited a logarithmic dependence.

The complex tumor flow phantom was designed to allow combined evaluation of both tissue-bound SPIONs and SPION suspensions in flow. This phantom features a carotid bifurcation flow channel and an ellipsoid tumor inclusion containing 1 mg Fe/mL of SPIONs. A SPION suspension with the same concentration (1 mg Fe/mL) was circulated through the flow channel. The results of the MMUS imaging for this phantom are presented in three dimensions in Figure 6, and as two-dimensional slices in Figure 7, where comparisons between water and SPION suspensions in the flow channel are made.

In the 3D visualization (Figure 6), the tumor inclusion is clearly identified and marked with black points. Regions showing no displacement correspond to the tumor inclusion and the flow channels, where MMUS fails to detect SPIONs in flow. When comparing the 3D displacement maps for water and SPION-filled flow channels, additional displacement is observed anterior to and adjacent to the right flow channel in the presence of SPIONs. These displacements occur below the flow channel and cannot be easily distinguished from those arising from tissue regions containing SPIONs.

Figure 7 illustrates this effect in two cross-sections along the y-axis. One cross-section contains only the flow channel without surrounding SPION-loaded tissue, while the other includes the region beyond the bifurcation where SPION-loaded tissue is present. As seen in the standard flow phantom experiments, the presence of SPIONs in the flow channel alone results in a displacement artifact located below the channel. Interestingly, when SPIONs are present both within the flow channel and in the surrounding tissue, their combined magnetic response leads to a superposition of displacement. This effect becomes evident when comparing the MMUS displacement maps for water-filled and SPION-filled flow channels.

The three-dimensional PCM results are presented in Figure 8 for both water and a 1 mg Fe/mL SPION suspension circulating in the flow channel. A FUS field with a peak negative pressure of 0.65 MPa is applied to the phantom, which induces acoustic cavitation in the presence of SPIONs. The resulting acoustic emissions are detected by a linear array transducer. This measurement is repeated at multiple positions along the *y*-axis.

Figure 8 demonstrates that the flow structure becomes visible when PCM is used and SPIONs are present in the circulating fluid. The tumor region, although loaded with SPIONs, does not exhibit cavitation activity and, therefore, is not visible in the PCM map. When comparing results between water and SPION suspensions in the flow channel, only minimal cavitation activity is observed in the water case, likely due to the presence of residual air bubbles in the system. However, the cavitation activity level is a magnitude lower for water than for SPIONs.

## 4. Discussion

In this study we introduced a novel combination of magnetomotive ultrasound and passive cavitation mapping for dual imaging of SPIONs in tissue and in flow environments.

### 4.1. Ultrasound Phantom Fabrication

We developed and fabricated several types of ultrasound phantoms for evaluation, with particular emphasis on the complex tumor flow phantom that incorporates a carotid bifurcation and a tumor inclusion with SPIONs. This improved model provides a valuable platform for testing SPION-targeted ultrasound imaging technologies under conditions that resemble realistic flow and tissue environments. In future work, the flow characteristics within the phantom should be characterized in greater detail, for example by combining experimental data with simulations. It will also be important to include a model of the cardiac cycle, capturing both diastolic and systolic phases, to more accurately replicate physiological conditions.

### 4.2. Magnetomotive Ultrasound

We implemented an enhanced MMUS method that combines frequency- and phase-sensitive detection with a global ultrasound elastography framework. This approach enabled visualization of SPION distributions with an iron content of 0.5 mg Fe/mL at a depth of 20 mm from the magnetic pole tip. The method provided reliable estimation of SPION-induced displacements and builds on earlier validation of GLUE for magnetic nanoparticle tracking [67]. Our findings extend the applicability of this technique and demonstrate its robustness across different phantom models used in this study.

In this work a magnetic field of 1 Hz was applied, determined by the characteristics of the MDT electromagnet. The system is designed to generate a large magnetic flux density (1 T at the pole tip) and a strong field gradient, but not for rapid modulation. As a DC magnet with high inductance, its field can only be varied at very low frequencies due to hysteresis effects, magnetic relaxation, and the large inductance of the coil. Consequently, in our experiments the operating frequency was limited to 1 Hz. This low frequency can lead to decorrelation effects and also hinders the real time application of this modality, however no additional hardware is required besides the already utilized MDT-Magnet.

The results indicate that frequency- and phase-sensitive global MMUS primarily measures the displacement of elastic tissue rather than the direct motion of SPIONs. This limitation arises because fluid suspensions lack the elastic restoring forces required to return to their original position once the magnetic field is removed. As a result, SPION suspensions are drawn toward the magnet only during magnetic field application, creating pressure on adjacent tissue and potentially producing misleading displacement signals. Although displacements within the flow channel do occur, the acquisition time of 10 s and flow-related decorrelation between radiofrequency datasets prevent these motions from being reliably mapped, in addition to the absence of a consistent displacement pattern.

In contrast, in the tissue phantom SPIONs are embedded within the elastic lattice of the PVA matrix. When exposed to a magnetic field, they undergo displacement but subsequently return to their original position due to the elasticity of the matrix. This reversible motion enables accurate MMUS mapping. Therefore, when applying MMUS to biological systems it is essential to distinguish between tissue-bound and freely circulating particles, since the technique reliably reflects only the former.

### 4.3. Passive Cavitation Mapping

PCM was highly effective in detecting cavitation events from SPION suspensions in flow and confirmed the sonosensitive properties of our LA SPIONs. Cavitation activity was visible at concentrations as low as 0.1 mg Fe/mL, which demonstrates the high sensitivity of the method for monitoring circulating particles. Cavitation followed a roughly logarithmic trend over iron content.

The observed cavitation response can be attributed to two primary mechanisms. First, the hydrophobic surface of the LA-SPIONs promotes heterogeneous bubble nucleation when exposed to focused ultrasound, effectively lowering the energetic threshold for cavitation [79,80]. Second, residual gas pockets originating from nanoparticle synthesis act as pre-existing nuclei, further enhancing cavitation onset [81,82]. While these mechanisms enhance cavitation sensitivity at low iron concentrations, the cavitation activity does not continue to rise proportionally at higher concentrations. Instead, a sublinear, or approximately logarithmic dependence on iron content is observed (see Figure 5f).

This nonlinear scaling arises because, beyond a certain particle density, increasing SPION concentration no longer increases the number of active cavitation sites. Three effects contribute to this saturation behavior: (i) the microbubbles that form during cavitation scatter and absorb part of the incident and re-emitted acoustic energy, leading to attenuation during both transmit and receive (self-shielding of the acoustic field) [83,84], (ii) strong bubble–bubble interaction [85] and bubble shielding [86] within dense bubble clusters reduces the oscillation amplitude of individual bubbles and (iii) the residual gas pockets are empty. As a consequence, further increases in nanoparticle concentration yield diminishing increments in cavitation intensity, resulting in the observed logarithmic trend.

PCM has limitations because cavitation intensity depends strongly on tissue properties such as density, speed of sound, reflection, and temperature. For this reason absolute values cannot be compared between different tissue conditions. PCM is, therefore, best used for relative assessments such as identifying active regions or tracking temporal changes during therapy.

In this study inertial cavitation was applied for imaging. This process can improve drug delivery by increasing vascular permeability and triggering drug release, but it also produces strong thermal and mechanical effects that may harm healthy vasculature. Stable cavitation is considered safer and can still be used both for imaging and to improve tumor permeability. Cavitation, therefore, has a dual role that can be beneficial for therapy while also requiring careful control to avoid unwanted side effects.

### 4.4. Advantages, Limitations and Challenges of the Dual Modality Approach

Frequency and phase sensitive gMMUS proved highly effective for visualizing tissue bound SPIONs, generating clear and quantifiable displacement maps. In contrast, when SPIONs circulated in suspension, MMUS produced misleading displacement signals beneath vascular structures. This reflects a key limitation, since the method depends on elastic restoring forces that are present in solid tissue but absent in fluid environments.

The complementary strengths of MMUS and PCM were evident in the phantom experiments. MMUS accurately identified tissue embedded SPIONs, while PCM reliably detected SPIONs in circulation down to very low concentrations. In regions where tissue and flow coexisted, MMUS alone produced superimposed displacement patterns that were difficult to interpret. PCM provided the necessary context to resolve these ambiguities, demonstrating the added value of integrating both modalities.

This dual modality framework offers several advantages. It enables simultaneous tracking of SPIONs in both tissue and vascular structures, which is essential for understanding accumulation, retention, and clearance during MDT. It also leverages widely available ultrasound systems, making it potentially translatable to clinical settings.

At the same time, important limitations remain. MMUS sensitivity is reduced in highly dynamic flow conditions, while PCM signals are strongly affected by tissue specific acoustic properties such as density, speed of sound, attenuation, and reflection. This variability complicates the extraction of absolute quantitative values, restricting PCM to relative comparisons. Furthermore, inertial cavitation used in PCM can promote drug delivery in tumors but may pose risks in healthy vasculature, requiring careful control of exposure parameters.

A practical implication for future MDT applications is that PCM can be used not only for detecting cavitation, but also for regulating it in real time. By adjusting MI, burst length, PRF, duty cycle, and exposure time based on the observed cavitation intensity, exposure can be maintained in a stable regime in healthy vasculature while allowing controlled inertial cavitation only at the tumor site. This establishes PCM as both a monitoring and feedback-safety mechanism during ultrasound-assisted MDT similar to the use in blood–brain-barrier opening [87].

The main challenge moving forward is to optimize the integration of these two modalities for in vivo use. This includes improving acquisition speed, mitigating motion artefacts, and establishing robust thresholds for cavitation detection under heterogeneous biological conditions. If addressed, the dual approach could provide a powerful tool for real time theranostic monitoring of SPION distribution during preclinical and clinical magnetic drug targeting.

### 4.5. Future Work

While the present study focused on ultrasound-based assessment of SPION distribution rather than full flow-field characterization, future work will incorporate external validation of hemodynamics. In particular, synchrotron X-ray PIV provides a direct quantitative reference for Reynolds number, wall shear stress, and multiphase velocity fields, and would enable benchmarking of the phantom flow regime against physiological conditions. Integrating X-ray PIV as a complementary modality would therefore strengthen the link between imaging performance and realistic vascular flow.

In addition, future work will include quantitative validation of the fusion consistency between MMUS and PCM. This requires fiducial markers at the tumour–vascular interface and a dedicated co-registration pipeline to enable direct comparison of the two modalities [88,89]. Such a setup would allow Bland–Altman or landmark-based agreement analysis to quantify fusion error, which was beyond the scope of the present feasibility study but represents the logical next step toward in vivo translation.

Future studies should also broaden the evaluation of the global MMUS method across different experimental conditions including in vitro, ex vivo, and in vivo models. Comparisons with existing MMUS implementations and simulations will help establish its robustness and limitations. The current two dimensional framework can be extended to full three dimensional imaging, which the GLUE approach is capable of supporting. This development is expected to improve sensitivity, spatial resolution, and accuracy of SPION localization. Incorporating not only axial displacement but also lateral and elevational components would provide a more complete mechanical contrast and allow more precise mapping of nanoparticle distribution.

For PCM, a key priority is to optimize SPION design in order to improve cavitation sensitivity and reproducibility. Strategies may include refining synthesis protocols, embedding SPIONs in micro- or nanobubbles, or combining them with gas generating agents. These modifications could enhance both imaging contrast and therapeutic efficiency. In addition, in vivo studies are needed to assess the role of PCM not only as an imaging tool but also as a therapeutic enhancer. Cavitation driven processes such as increased vascular permeability or triggered local drug release hold strong promise but require careful control to avoid harmful effects in healthy tissue.

A practical obstacle for translation is the geometric and mechanical integration of all required modalities into a single platform. In a clinical setting, the MDT magnet must be in direct contact with the patient in order to generate a sufficiently steep field gradient, while both the FUS transducer and the diagnostic ultrasound array require continuous acoustic coupling via gel or a water pathway. This creates strict spatial constraints at the patient interface, particularly when the anatomical access window is small or curved. Moreover, actuator kinematics become non-trivial: the electromagnet in the animal operation room is currently mounted on a 6-axis robotic arm, and a future integrated system would either require the ultrasound transducers to be co-mounted, which limits the available incidence angles, or the addition of a second synchronized robotic positioning stage. As a consequence, collision-free positioning, acoustic coupling stability, and alignment between magnetic actuation and acoustic sensing all become coupled engineering problems that must be solved jointly in future platform designs.

## 5. Conclusions

This study demonstrated a dual modality approach combining magnetomotive ultrasound and passive cavitation mapping for imaging SPIONs in both tissue and flow. MMUS effectively visualized tissue bound particles, while PCM sensitively detected circulating suspensions. Together they provide complementary information for accurate localization. Although challenges remain for in vivo use, such as hardware optimization and reliable image interpretation, this strategy offers a promising foundation for advanced ultrasound based theranostic systems in cancer therapy.

## Figures and Tables

**Figure 1 sensors-25-07171-f001:**
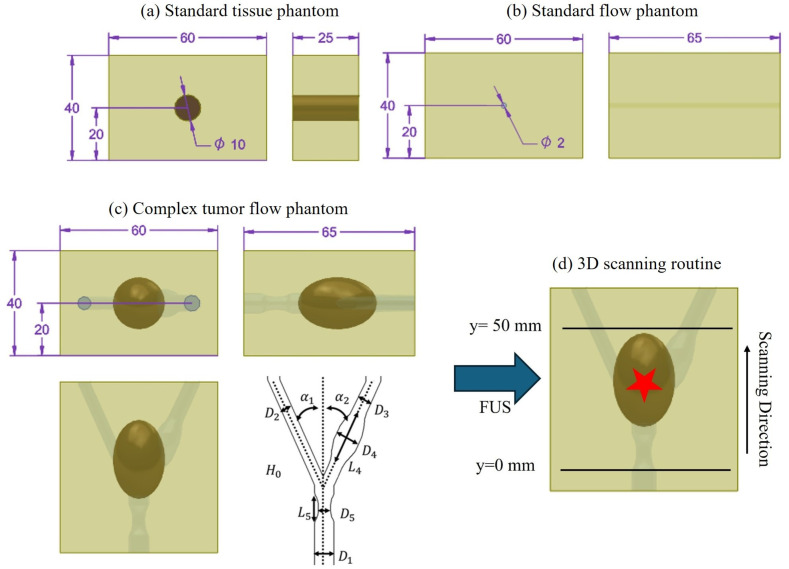
Models of the used ultrasound phantoms (measurements in mm) for (**a**) standard tissue phantom, (**b**) standard flow phantom, and (**c**) complex flow tumor phantom. In (**c**), the sketch for the bifurcation is shown ($H_{0}$ = 100 mm, $\alpha_{1}$ = $\alpha_{2}$ = 15°, $D_{1}$ = 8 mm, $D_{2}$ = 4.6 mm, $D_{3}$ = 5.8 mm, $D_{4}$ = 10 mm, $L_{4}$ = 20 mm, $D_{5}$ = 5 mm, $L_{5}$ = 12 mm). The tumor is an ellipsoid with a diameter of 20 mm and a length of 30 mm. (**d**) illustrates the 3D scanning routine, in which the position of the magnetic pole tip is directly below the red star.

**Figure 2 sensors-25-07171-f002:**
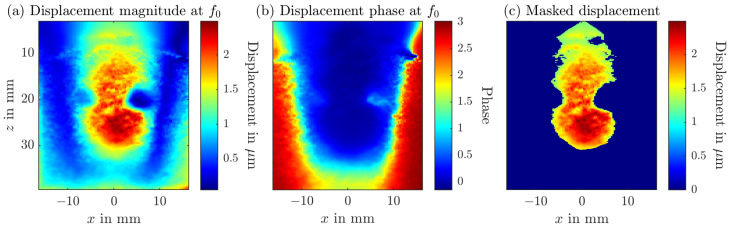
Principle of frequency- and phase-sensitive global magnetomotive ultrasound, illustrated using the complex flow tumor phantom. (**a**) Displacement amplitude at the magnetic excitation frequency $f_{0}$. (**b**) Corresponding phase map used as a mask. (**c**) Final result after applying the phase mask to isolate SPION-induced displacements.

**Figure 3 sensors-25-07171-f003:**
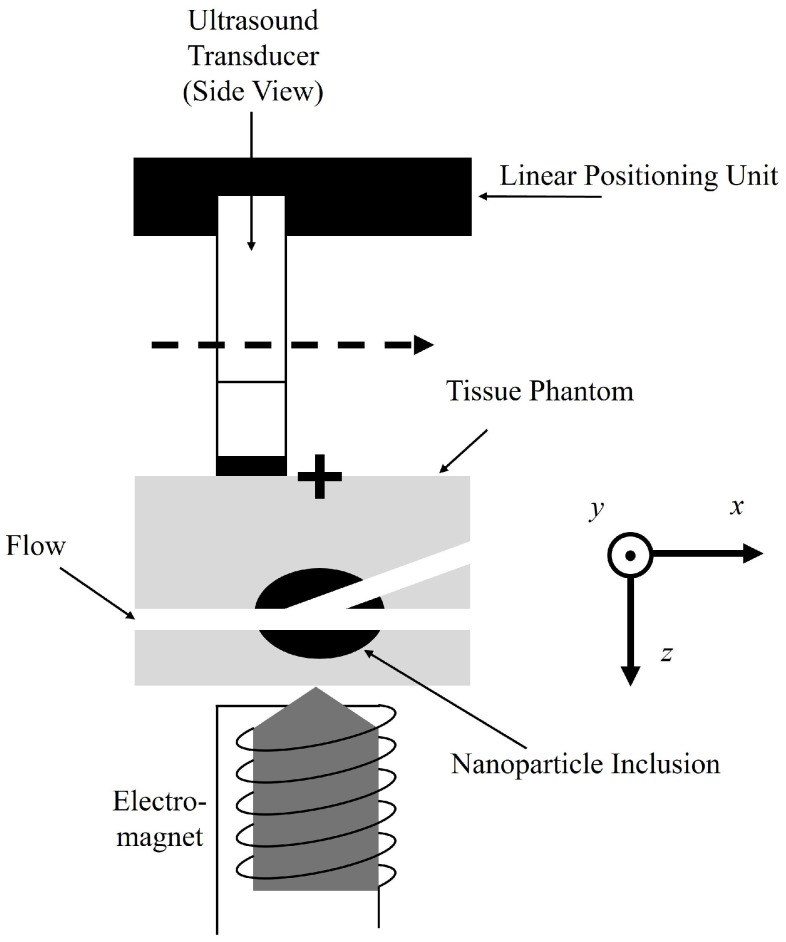
Schematic of the measurement setup used for magnetomotive ultrasound (MMUS). The + shows the origin of the utilized coordinate system and is directly in line with the center of the magnetic pole tip (x,y=0) and starts and z=0 is directly above the ultrasound phantom. For the different MMUS evaluations, the flow phantom will be switched with the corresponding standard phantom.

**Figure 4 sensors-25-07171-f004:**
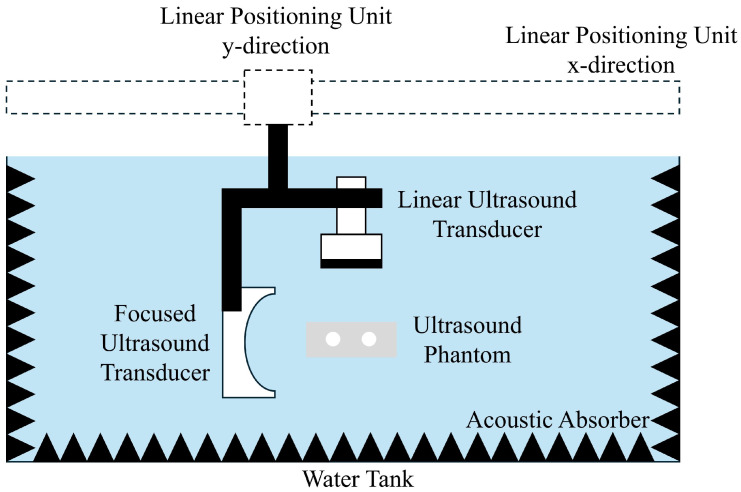
Measurement setup used for passive cavitation mapping of superparamagnetic iron oxide nanoparticles (SPIONs). The focused ultrasound (FUS) transducer is placed on the depth z=20 mm and focuses its field on the ultrasound phantom. The linear array transducer is placed right above the phantom at z=−20 mm. The FUS and linear array are fixed on a structure with is again fixed on a 2D linear positioning unit.

**Figure 5 sensors-25-07171-f005:**
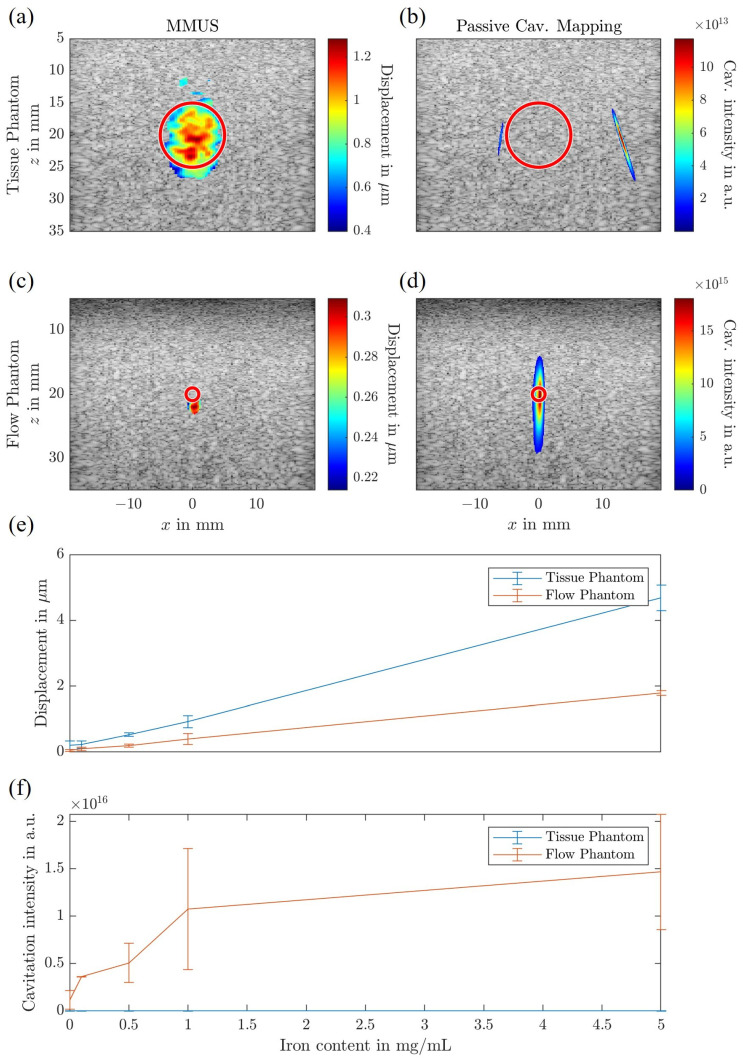
Results of the frequency- and phase-sensitive global magnetomotive ultrasound (MMUS) measurements and passive cavitation mapping (PCM) for both the tissue and flow phantoms. (**a**,**b**) show the MMUS and PCM results for the tissue phantom, respectively, while (**c**,**d**) present the corresponding MMUS and PCM results for the flow phantom. (**e**) illustrates the maximum measured MMUS displacement over iron content, and (**f**) shows the cavitation activity over iron content for the flow phantom (orange) and the tissue phantom (blue).

**Figure 6 sensors-25-07171-f006:**
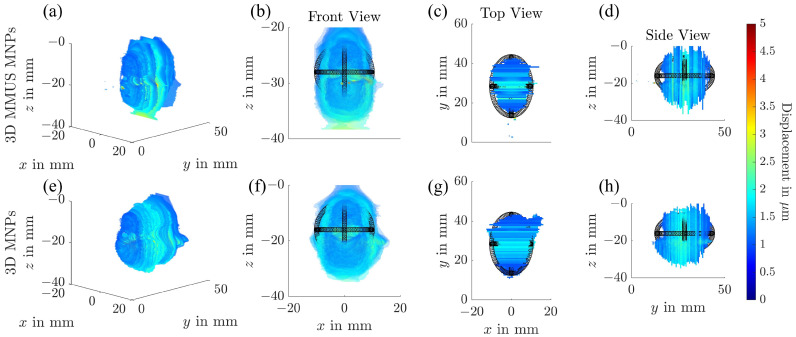
Three-dimensional MMUS reconstructions obtained using frequency- and phase-sensitive magnetomotive ultrasound. (**a**–**d**) show the measurements acquired for water in the flow channel, while (**e**–**h**) display the results for LA-SPIONs. The additional tumor region containing the SPIONs is indicated by the black dots. (**a**,**e**) present an isometric view, (**b**,**f**) a front view, (**c**,**g**) a top view, and (**d**,**h**) a side view of the reconstructed displacement field.

**Figure 7 sensors-25-07171-f007:**
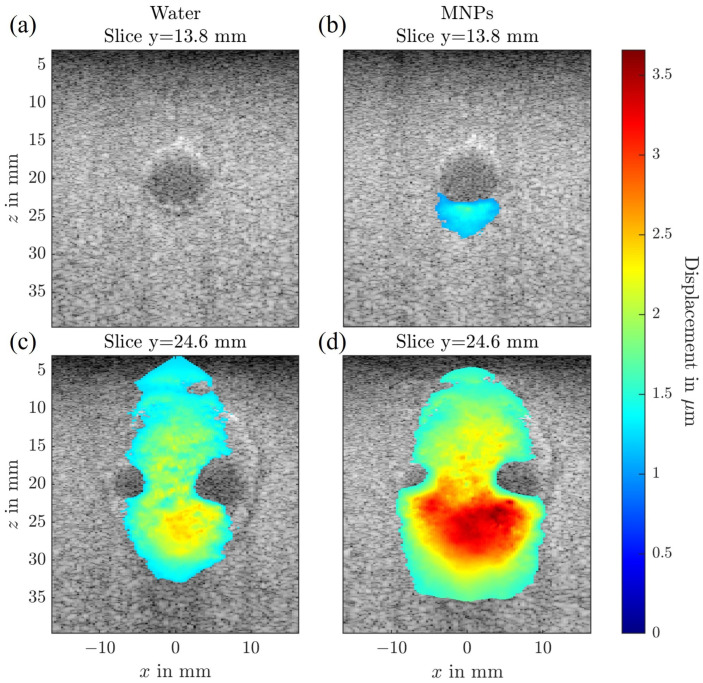
Frequency- and phase-sensitive global magnetomotive ultrasound (MMUS) results for the complex flow tumor phantom overlaid over the corresponding b-mode image. (**a**,**c**) show measurements with water circulating through the flow channels, whereas (**b**,**d**) depict measurements with a 1 mg Fe/mL LA-SPION suspension. (**a**,**b**) correspond to a slice at y=13.8 mm, where only a single flow channel is present, while (**c**,**d**) show a slice at y=24.6 mm that includes a bifurcated flow channel surrounded by the LA-SPION-laden tumor structure.

**Figure 8 sensors-25-07171-f008:**
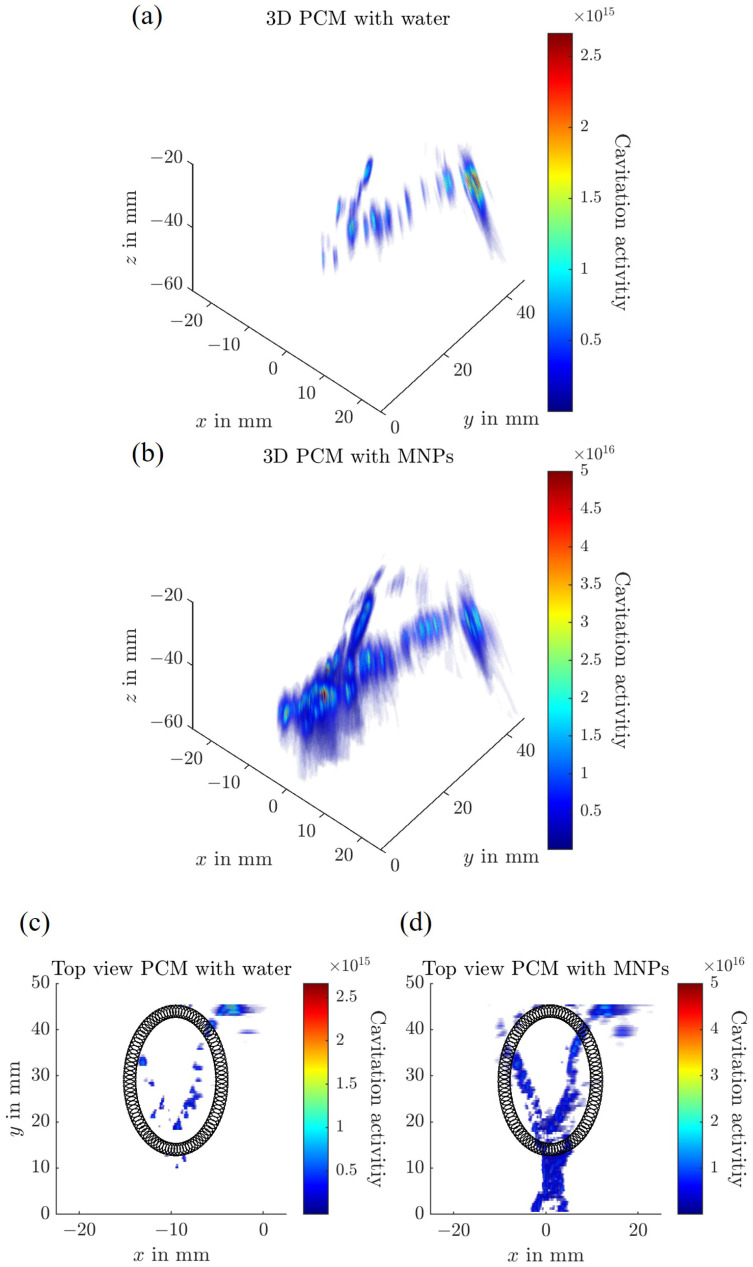
Three-dimensional passive cavitation mapping (PCM) of the complex flow tumor phantom. (**a**,**c**) show the PCM results with water circulating through the flow channels, whereas (**b**,**d**) depict the measurements obtained with a 1 mg Fe/mL LA-SPION suspension. (**a**,**b**) present isometric views of the reconstructed cavitation activity, while (**c**,**d**) show the corresponding top-view projections. The tumor region is outlined by the black dots in the top-view images.

## Data Availability

The data are available upon request to the first author.

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
