# Peer review of "Dual-Modality Ultrasound Imaging of SPIONs Distribution via Combined Magnetomotive and Passive Cavitation Imaging"

_sensors, 2025, doi:10.3390/s25237171_

Round 1

Reviewer 1 Report

Comments and Suggestions for Authors

The manuscript “Combined Magnetomotive and Passive Cavitation Imaging within Complex Ultrasound Flow Phantom” describes a series of proof-of-concept experiments demonstrating the complementarity of two different ultrasound imaging modalities, the first relying on magnetomotive imaging of magnetic nanoparticles, and the second using passive cavitation mapping. The concept is interesting, and the potential for biomedical applications, particularly using magnetic drug delivery and/or magnetic particle imaging should motivate future research interest. The authors are reasonably measured when discussing the potential of the combined techniques, while also addressing limitations and engineering challenges that will need to be overcome on the path towards clinical translation. However, aspects of the manuscript will need to be significantly improved prior to publication. I have concerns about the level of experimental detail and the clarity of the presentation of important results. Could the authors address the following points for their revised manuscript?

  1. More information needs to be provided about the characterization of the SPIONs. Do the authors have data on the morphology of the nanoparticles? Size, shape and polydispersity of the iron oxide core, and shell thickness? Have the authors characterized the iron oxide phase, or the magnetic properties, particularly the superparamagnetic blocking temperature or the low-field susceptibility?
  2. Little information is provided about the electromagnet field source, except to say that the field from the pole tip was 1 T. To my understanding, the region of interest associated with the tumor in the flow phantom is ~20 mm from the pole tip, so the field at that region would be significantly less. Given that the magnetomotive force depends on the field gradient, has the field profile been characterized? Was the magnet bespoke or acquired commercially, and, if the latter, can the authors provide a model and manufacturer?
  3. The main results are displayed in Figs 5-8, but all these figures need to be improved from a presentation and clarity perspective. Please include subfigure labels in each of these figures. Please rewrite the captions to refer to these labels, so it is clear what is being described in each figure. Please write more descriptive captions to make it easier for the reader to follow.
  4. In Section 3, page 11, is the legend in the Cavitation intensity graph in Figure 5 mislabeled? This seems to be inconsistent with the text in paragraph 1 of page 12, where it is noted that no cavitation was observed within the tissue phantom?
  5. On page 13, paragraph 1, the authors point out that the relationship between iron content and cavitation activity exhibited a logarithmic trend. Do the authors have an explanation as to why this might be the case?
  6. In Section 4.5, page 17, paragraph 3, the authors discuss the challenge of integrating MMUS and PCM into a unified platform. I’m wondering if the authors would be prepared to briefly discuss some of engineering challenges they can envisage, particularly in terms of device integration and the geometric challenges of combining multiple excitation and imaging modalities into a single device or platform.
  7. Please provide an updated Acknowledgments section.
  8. Some very minor grammatical errors: In the introduction, on page 3, “In simplified scenarios, the MMUS displacement can also be used to estimated the SPION concentration” should be “[…] used to estimate […]”. In the captions for Figure 4, 6 and 8, “iron oxide” is two words.

Author Response

Comment 1: 

More information needs to be provided about the characterization of the SPIONs. Do the authors have data on the morphology of the nanoparticles? Size, shape and polydispersity of the iron oxide core, and shell thickness? Have the authors characterized the iron oxide phase, or the magnetic properties, particularly the superparamagnetic blocking temperature or the low-field susceptibility?

Reply 1:

The nanoparticles used in this work are lauric-acid stabilized superparamagnetic iron oxide nanoparticles (LA-SPIONs) synthesized according to the established SEON protocol routinely used in our group [20, 48]. The same synthesis route and formulation have been extensively characterized in earlier studies, which include TEM analysis of the iron oxide cores, DLS analysis of the hydrodynamic size and polydispersity, FTIR confirmation of the surfactant coating, and SQUID/VSM measurements of the magnetic behavior. These previously published data therefore directly apply to the material used in the present study.

As the aim of the present work is not the development of a new nanoparticle formulation, but rather the demonstration of a dual-modality ultrasound imaging strategy, no additional TEM/DLS/VSM measurements were repeated here; instead, we relied on this already validated and published nanoparticle platform. To make this clearer to the reader, we added a section to the method section and are referring explicitly to the corresponding characterization literature.

We added the subsection “Superparamagnetic Iron Oxide Nanoparticles” in the Materials and Methods section:

“The nanoparticles used in this work are lauric-acid coated superparamagnetic iron oxide nanoparticles (LA-SPIONs), synthesized according to the protocol from [20,48]. This formulation has been extensively characterized in earlier studies, and the reported physicochemical and magnetic properties directly apply to the material used here.

Transmission electron microscopy measurements in [48] show that the magnetite cores have a diameter of approximately 7–8 nm, forming multicore aggregates embedded in a lauric-acid matrix. Dynamic light scattering measurements of the same formulation report a hydrodynamic diameter on the order of 100 nm with low polydispersity. Magnetic characterization using vibrating sample magnetometry and superconducting quantum interference device susceptometry demonstrates superparamagnetic behavior at room temperature with negligible hysteresis and a blocking temperature below physiological conditions [20]. The saturation magnetization is reported to be approximately 450 kA/m for this formulation, and the absence of remanent magnetization outside an applied magnetic field is confirmed, which is essential for biomedical applications.”

Comment 2:

Little information is provided about the electromagnet field source, except to say that the field from the pole tip was 1 T. To my understanding, the region of interest associated with the tumor in the flow phantom is ~20 mm from the pole tip, so the field at that region would be significantly less. Given that the magnetomotive force depends on the field gradient, has the field profile been characterized? Was the magnet bespoke or acquired commercially, and, if the latter, can the authors provide a model and manufacturer?

Reply 2:

We thank the reviewer for this comment. The electromagnet used in this study is a custom-built high-gradient MDT unit developed in collaboration with Siemens AG specifically for magnetic drug targeting applications. It is not commercially available and was engineered to generate a very strong and spatially confined magnetic field directly at the pole tip. We agree that the field strength decreases rapidly with distance from the pole surface and that the gradient, not the static field amplitude, is the decisive quantity for magnetomotive displacement. To address this more clearly, we have now added a reference to our previously published characterization of this magnet, where the axial field decay and gradient distribution were experimentally measured and validated against simulation (references [20, 66] in the revised manuscript). In addition, we now provide a new corresponding COMSOL field simulation in the Supplementary Materials (figure S1 and S2), including the field profile along the z-axis between the pole tip and the tumor region of the phantom.

As shown in the new supplementary plot, the magnetic flux density decreases from 1 T at the pole tip to values in the order of 400 mT at the phantom depth used in this work, but the gradient remains sufficiently high to induce measurable magnetomotive forces in tissue-bound SPIONs. This is consistent with previous MDT studies using the same electromagnet design [20] and explains why displacement amplitudes decrease with distance from the magnet while still remaining detectable in the presented frequency- and phase-sensitive MMUS reconstructions.

For clarity, we have also added a sentence explicitly noting that the magnet is a bespoke system developed by Siemens AG and is not commercially distributed:

“Below the phantom the custom made MDT electromagnet (Siemens AG) is placed [20,66] (see Figure S1 and S2 for further information), which induces a magnetic field with a sinusoidal time function with an offset of half the amplitude to only have positive values. The frequency of the applied magnetic field is f0 = 1 Hz and has a maximum magnetic flux density of approx. B = 1 T directly on the pole tip and approx. 0.4 T at z = 20 mm. The respective magnetic flux density gradient Bz/z is around 90 T/m and 15 T/m.”

Comment 3:

The main results are displayed in Figs 5-8, but all these figures need to be improved from a presentation and clarity perspective. Please include subfigure labels in each of these figures. Please rewrite the captions to refer to these labels, so it is clear what is being described in each figure. Please write more descriptive captions to make it easier for the reader to follow.

Reply 3:

In the revised manuscript, all four figures (Figs. 5–8) have been updated to include clearly visible subfigure labels. The figure captions were rewritten accordingly so that each subpanel is explicitly referenced in the text, making it easier for the reader to associate the visual content with its interpretation. We have also expanded the captions to provide more descriptive context for the imaging results.

Comment 4:

In Section 3, page 11, is the legend in the Cavitation intensity graph in Figure 5 mislabeled? This seems to be inconsistent with the text in paragraph 1 of page 12, where it is noted that no cavitation was observed within the tissue phantom?

Reply 4:

Thank you for pointing this out. The reviewer is correct that cavitation was not observed in the tissue phantom, and the original legend of Figure 5 was unintentionally misleading in this respect. The label has now been corrected to clearly indicate that cavitation activity is shown only for the flow phantom, while the tissue phantom curve represents baseline noise.

Comment 5:

On page 13, paragraph 1, the authors point out that the relationship between iron content and cavitation activity exhibited a logarithmic trend. Do the authors have an explanation as to why this might be the case?

Reply 5:

We added a paragraph in the discussion section:

The observed cavitation response can be attributed to two primary mechanisms. First, the hydrophobic surface of the LA-SPIONs promotes heterogeneous bubble nucleation when exposed to focused ultrasound, effectively lowering the energetic threshold for cavitation [74,75]. Second, residual gas pockets originating from nanoparticle synthesis act as pre-existing nuclei, further enhancing cavitation onset [76,77]. While these mechanisms enhance cavitation sensitivity at low iron concentrations, the cavitation activity does not continue to rise proportionally at higher concentrations. Instead, a sublinear, or approximately logarithmic dependence on iron content is observed (see Figure 5 f).

This nonlinear scaling arises because, beyond a certain particle density, increasing SPION concentration no longer increases the number of active cavitation sites. Three effects contribute to this saturation behavior: (i) the microbubbles that form during cavitation scatter and absorb part of the incident and re-emitted acoustic energy, leading to attenuation during both transmit and receive (self-shielding of the acoustic field) [78,79], (ii) strong bubble–bubble interaction [80] and bubble shielding [81] within dense bubble clusters reduces the oscillation amplitude of individual bubbles and (iii) the residual gas pockets are empty. As a consequence, further increases in nanoparticle concentration yield diminishing increments in cavitation intensity, resulting in the observed logarithmic trend.

Comment 6:

In Section 4.5, page 17, paragraph 3, the authors discuss the challenge of integrating MMUS and PCM into a unified platform. I’m wondering if the authors would be prepared to briefly discuss some of engineering challenges they can envisage, particularly in terms of device integration and the geometric challenges of combining multiple excitation and imaging modalities into a single device or platform.

Relpy 6:

We agree with the reviewer that the integration of MMUS and PCM into a single clinical device is not only technically challenging but also a central engineering bottleneck for future translation. In the revised manuscript, we have now expanded the discussion in Section 4.5 accordingly. In particular, the main difficulty arises from the fact that three separate hardware components must be physically co-aligned at the patient interface: the MDT electromagnet, the focused ultrasound (FUS) transducer for cavitation excitation, and the diagnostic ultrasound array for MMUS/PCM signal detection. While the electromagnet must be in direct contact (or close as possible) with the body to generate a sufficiently steep gradient field, both ultrasound transducers require continuous acoustic coupling through gel or a water path, which creates strict geometric constraints in already limited anatomical access regions. This becomes especially critical for tumors located in narrow or curved anatomical windows.

A further challenge concerns device kinematics. At present, the electromagnet is mounted on a 6-axis robotic positioning arm (Stäubli TS200). For a combined theranostic system, either the imaging transducer would need to be co-mounted on the same arm—which introduces constraints on the relative acoustic incidence angle and coupling—or an additional independent but synchronized robotic positioning stage would be required for the acoustic components. This imposes non-trivial requirements on collision-free path planning, field-of-view stability, and mutual alignment of excitation and sensing geometries in real time. These constraints illustrate why a unified platform remains a long-term engineering goal and not yet an off-the-shelf solution.

We added the following paragraph to the discussion section:

“A practical obstacle for translation is the geometric and mechanical integration of all required modalities into a single platform. In a clinical setting, the MDT magnet must be in direct contact with the patient in order to generate a sufficiently steep field gradient, while both the FUS transducer and the diagnostic ultrasound array require continuous acoustic coupling via gel or a water pathway. This creates strict spatial constraints at the patient interface, particularly when the anatomical access window is small or curved. Moreover, actuator kinematics become non-trivial: the electromagnet in the animal operation room is currently mounted on a 6-axis robotic arm, and a future integrated system would either require the ultrasound transducers to be co-mounted, which limits the available incidence angles, or the addition of a second synchronized robotic positioning stage. As a consequence, collision-free positioning, acoustic coupling stability, and alignment between magnetic actuation and acoustic sensing all become coupled engineering problems that must be solved jointly in future platform designs.”

Comment 7:

Please provide an updated Acknowledgments section.

Reply 7:

Thank you. The Acknowledgments was updated.

Comment 8:

Some very minor grammatical errors: In the introduction, on page 3, “In simplified scenarios, the MMUS displacement can also be used to estimated the SPION concentration” should be “[…] used to estimate […]”. In the captions for Figure 4, 6 and 8, “iron oxide” is two words.

Reply 8:

Thank you. This was corrected.

Reviewer 2 Report

Comments and Suggestions for Authors

The authors present a very interesting investigation of combined magnetomotive and passive cavitation imaging. The manuscript is suitable for publication in this journal after the authors provide additional clarification regarding Figure 5:

Can it be stated that "The MMUS displacement demonstrates a clear linear dependence on iron content" given the limited number of data points?

Is the observed cavitation activity truly linear with respect to iron content in the range of 1 to 5 mg/mL?

Author Response

Comment 1:

Can it be stated that "The MMUS displacement demonstrates a clear linear dependence on iron content" given the limited number of data points?

Reply 1:

We thank the reviewer for this helpful remark. The corresponding statement in the manuscript has been revised accordingly. While the measured displacement increases consistently with rising iron content, the limited number of tested concentrations does not justify a definitive quantitative claim of strict linearity. We now describe the trend as “increasing approximately proportionally with iron content within the tested concentration range.”

We would also like to clarify that such a proportional relationship is physically expected in the displacement regime relevant to MMUS. At micrometer-scale deformations, as observed here, the surrounding material behaves effectively linearly elastic, and the magnetomotive force scales linearly with the magnetic susceptibility and therefore with iron content. Nonlinear mechanical effects would only be expected at much larger displacements (e.g., in the millimeter range), where strain stiffening or geometric nonlinearities may occur. Within the displacement amplitudes characteristic of magnetic nanoparticle tracking, a near-linear dependence is therefore both theoretically justified and experimentally consistent. The mechanical properties of the phantom or tissue act as a proportionality factor in this regime and do not alter the underlying linear scaling.

Comment 2:

Is the observed cavitation activity truly linear with respect to iron content in the range of 1 to 5 mg/mL?

Reply 2:

The cavitation activity is indeed not linear with respect to iron content. As correctly noted, the manuscript already describes a logarithmic trend. This sublinear behavior is expected from a physical perspective, because with increasing SPION concentration, cavitation nuclei begin to self-shield the acoustic field and bubble–bubble interactions reduce the effective oscillation amplitude. As a result, the growth of the received PCM signal saturates rather than continuing proportionally. The text in the Results section has been slightly clarified to make this interpretation more explicit.

Reviewer 3 Report

Comments and Suggestions for Authors

This manuscript proposes a dual-modality technique combining magnetodynamic ultrasound (MMUS) and passive cavitation imaging (PCM) to monitor the distribution of superparamagnetic iron oxide nanoparticles (SPIONs) in complex vascular environments for optimizing magnetically targeted tumor therapy (MDT). The core of the study demonstrates that MMUS can localize SPIONs within tissue, while PCM can detect SPION cavitation signals in blood flow, complementing each other to overcome the limitations of a single technique. Experiments using a standard phantom and a 3D-printed carotid bifurcation tumor flow field phantom confirmed the effectiveness of this technique for static and dynamic SPION imaging. The manuscript innovatively combines dual modalities to achieve full-scene SPION monitoring, is compatible with existing clinical equipment, and has translational potential. However, the manuscript's limitations are limited to phantom experiments, lack of in vivo validation, failure to quantify particle concentration, and failure to explore potential artifacts.    

The content of the manuscript is within the scope of the journal and can be of broad interest to readers. However, in terms of specific content, there is still room for improvement. Therefore, I decided to make the decision of major revision. It is recommended that the author properly absorb the reviewers' comments and make corresponding improvements and enhancements.

1. The manuscript uses PCM to monitor acoustic cavitation, but lacks a detailed description of the fundamental nature of cavitation phenomena (e.g., bubble dynamics and energy distribution) that are common across acoustic and fluidic regimes. The authors are encouraged to reference relevant cavitation suppression strategies (e.g., controlling inertial cavitation to minimize tissue damage, see Ultrasonics Sonochemistry 86 (2022): 106035). This is relevant to the potential challenges of PCM applications, such as when discussing the potential for non-target biological effects of cavitation events or optimizing the acoustic response of SPIONs. The manuscript currently acknowledges the application risks of cavitation (e.g., the "biochemical and mechanical effects" mentioned on page 3) but does not delve into how to actively suppress cavitation.

Further explanation is recommended regarding the safety issues of balancing cavitation for imaging and therapy in MDT. In PCM experiments, reference should be made to suppression mechanisms to optimize ultrasound parameters (e.g., pressure amplitude) to avoid noise interference. Furthermore, to enhance the paper logic, the abstract on page 1, which emphasizes "complementary MMUS and PCM imaging," should include specific support for cavitation control.

2. The title and abstract are misaligned. The current title emphasizes "complex ultrasound hemodynamic modeling," but the actual innovation lies in the dual-modality imaging approach. We recommend revising the title to "Dual-modality Ultrasound Imaging of SPIONs Distribution via Combined Magnetomotive and Passive Cavitation Mapping."

3. The abstract does not highlight the methodological breakthrough. It is recommended to clarify and add: "This work achieves the first coordinated imaging of MMUS and PCM in the same anatomical model and time scale, resolving the key bottleneck of SPIONs monitoring in blood vessels/tissues."

4. Flow field dynamics parameters are not available. Key parameters such as the Reynolds number and wall shear force in the flow phantom are not specified, and additional pump speed-flow conversion formulas and physiological relevance verification are needed.

5. The manuscript relies on ultrasound to image SPIONs in the flow model, but ultrasound techniques may have resolution limitations or be unable to directly quantify density changes. A brief description of advanced flow field measurement tools (X-ray PIV) should be included, as its high resolution and multiphase flow capabilities can validate the models referenced in the literature, such as 'synchrotron X-ray based particle image velocimetry to measure multiphase streamflow and densitometry'. For example, in the complex carotid bifurcation model, X-ray PIV can be used to cross-validate hemodynamics and compensate for potential ultrasound errors. This would also enhance the scientific integrity of the manuscript. On page 2, the manuscript mentions "flow dynamics" modeling, but no external validation methods were used.
    It is recommended that relevant citations be added to the model validation section of the "Materials and Methods" section when describing the flow model. "The flow model was fabricated using 3D printing and PVA cryogel, and the flow characteristics were initially validated by ultrasound. To further improve accuracy, reference is made to X-ray PIV, which can quantify density and velocity distributions in multiphase flows." In the "Discussion" or "Future Work" sections, relevant descriptions should be added when highlighting model limitations or expanding applications. For example, "Current flow models rely on ultrasound to measure SPIONs concentration gradients, but their accuracy is limited by their resolution; combining them with advanced techniques such as synchrotron X-ray PIV could provide more reliable data and be applicable to vascular-like environments."

6. The highest concentration (5 mg Fe/mL) is outside the therapeutic window. Clinical MDT doses are typically ≤0.5 mg Fe/kg, requiring supplementation with physiologically relevant concentrations (0.01–0.1 mg/mL). The impact of high concentrations on the cavitation signal saturation and PCM quantitative accuracy should be discussed.

7. Fusion consistency has not been verified. A validation target should be set at the tumour-vascular interface to compare the localization error between single-modality and fusion imaging (Bland-Altman analysis is recommended).

8. Insufficient control experiments led to weaker evidence. Key control groups were missing, such as:
â–¶ Non-magnetic nanoparticles (e.g., liposomes): to verify the specificity of MMUS;
â–¶ Microbubble contrast agents: to rule out the possibility that the PCM signal is due to residual gas;
â–¶ Ex vivo tissue validation: to demonstrate that the phantom conclusions can be extrapolated to biological tissue.

Author Response

Comment 1:

The manuscript uses PCM to monitor acoustic cavitation, but lacks a detailed description of the fundamental nature of cavitation phenomena (e.g., bubble dynamics and energy distribution) that are common across acoustic and fluidic regimes. The authors are encouraged to reference relevant cavitation suppression strategies (e.g., controlling inertial cavitation to minimize tissue damage, see Ultrasonics Sonochemistry 86 (2022): 106035). This is relevant to the potential challenges of PCM applications, such as when discussing the potential for non-target biological effects of cavitation events or optimizing the acoustic response of SPIONs. The manuscript currently acknowledges the application risks of cavitation (e.g., the "biochemical and mechanical effects" mentioned on page 3) but does not delve into how to actively suppress cavitation.

Further explanation is recommended regarding the safety issues of balancing cavitation for imaging and therapy in MDT. In PCM experiments, reference should be made to suppression mechanisms to optimize ultrasound parameters (e.g., pressure amplitude) to avoid noise interference. Furthermore, to enhance the paper logic, the abstract on page 1, which emphasizes "complementary MMUS and PCM imaging," should include specific support for cavitation control.

Reply 1:

We thank the reviewer for this insightful comment. We agree that cavitation control and safety considerations are highly relevant when PCM is used in a theranostic context. In the revised manuscript, we have expanded the Discussion section to include a short explanation of how cavitation can be regulated during acoustic excitation (e.g., via MI, burst length, PRF, duty cycle and exposure time) in order to remain in the stable cavitation regime and prevent unwanted inertial effects in non-target regions.

We would like to note that the reference suggested by the reviewer (Ultrasonics Sonochemistry 86 (2022): 106035) addresses hydrodynamic cavitation suppression in fluidic flow systems. While conceptually related, this belongs to a different physical regime than ultrasound-driven cavitation in soft tissue. For this reason, we have instead cited biomedical ultrasound work specifically on acoustic cavitation control and nuclei management in vivo, which is more directly applicable to PCM-guided MDT.

In addition, we have added one sentence to the Abstract highlighting that the translational benefit of combining MMUS and PCM also includes the possibility of controlling cavitation activity via feedback, not only detecting it. Together, these revisions strengthen the logical link between imaging, therapy guidance, and safety.

Comment 2:

The title and abstract are misaligned. The current title emphasizes "complex ultrasound hemodynamic modeling," but the actual innovation lies in the dual-modality imaging approach. We recommend revising the title to "Dual-modality Ultrasound Imaging of SPIONs Distribution via Combined Magnetomotive and Passive Cavitation Mapping."

Reply 2:

We agree that the revised title better reflects the core contribution of the manuscript and have updated it accordingly

Comment 3:

The abstract does not highlight the methodological breakthrough. It is recommended to clarify and add: "This work achieves the first coordinated imaging of MMUS and PCM in the same anatomical model and time scale, resolving the key bottleneck of SPIONs monitoring in blood vessels/tissues."

Reply 3:

Thank you for this comment. We have updated the abstract to make the methodological contribution more explicit.

Comment 4:

Flow field dynamics parameters are not available. Key parameters such as the Reynolds number and wall shear force in the flow phantom are not specified, and additional pump speed-flow conversion formulas and physiological relevance verification are needed.

Reply 4:

We thank the reviewer for this valuable comment. The primary objective of this study was to demonstrate the feasibility of dual-modality SPION imaging using MMUS and PCM, rather than to perform a detailed fluid-dynamic characterization of the flow phantom. The flow system was driven by a peristaltic pump (Pharmacia LKB P-1) operating at a flow rate of approximately 5 mL/min through tubing with an inner diameter of 2.1 mm, providing a stable laminar flow regime. Since the pump tubing was connected to phantoms with varying internal channel geometries, accurate determination of Reynolds numbers and wall shear stresses requires dedicated simulation and flow analysis, which will be addressed in future work.

To clarify this point, we have added a brief statement in the Materials and Methods section specifying the pump type and flow rate, and an additional note in the Discussion/Future Work section indicating that future studies will include quantitative flow characterization, such as computational fluid dynamics or X-ray PIV, to determine Reynolds number and wall shear stress and to further validate the physiological relevance of the phantom setup. These revisions address the reviewer’s concern while keeping the focus of the present work on the imaging methodology rather than detailed hydrodynamic modeling.

Comment 5:

The manuscript relies on ultrasound to image SPIONs in the flow model, but ultrasound techniques may have resolution limitations or be unable to directly quantify density changes. A brief description of advanced flow field measurement tools (X-ray PIV) should be included, as its high resolution and multiphase flow capabilities can validate the models referenced in the literature, such as 'synchrotron X-ray based particle image velocimetry to measure multiphase streamflow and densitometry'. For example, in the complex carotid bifurcation model, X-ray PIV can be used to cross-validate hemodynamics and compensate for potential ultrasound errors. This would also enhance the scientific integrity of the manuscript. On page 2, the manuscript mentions "flow dynamics" modeling, but no external validation methods were used.
    It is recommended that relevant citations be added to the model validation section of the "Materials and Methods" section when describing the flow model. "The flow model was fabricated using 3D printing and PVA cryogel, and the flow characteristics were initially validated by ultrasound. To further improve accuracy, reference is made to X-ray PIV, which can quantify density and velocity distributions in multiphase flows." In the "Discussion" or "Future Work" sections, relevant descriptions should be added when highlighting model limitations or expanding applications. For example, "Current flow models rely on ultrasound to measure SPIONs concentration gradients, but their accuracy is limited by their resolution; combining them with advanced techniques such as synchrotron X-ray PIV could provide more reliable data and be applicable to vascular-like environments."

Reply 5:

We thank the reviewer for this suggestion. The intention of the present work was not to characterize the flow field itself, but to demonstrate that MMUS and PCM can jointly visualize tissue-bound versus circulating SPIONs in a realistic vascular geometry. For this reason, the ultrasound measurements in the flow model served as imaging validation, not as hydrodynamic ground-truthing. We agree, however, that advanced techniques such as synchrotron X-ray PIV or X-ray densitometry provide superior quantitative accuracy for hemodynamic validation and multiphase flow characterization.

We have therefore added a brief note in the Methods (at the end of Section 2.2.3) and an expansion in the Discussion/Future Work (at Section 4.5) to clarify that X-ray PIV would be an appropriate next step for external validation of the flow characteristics, particularly once the platform is extended toward in vivo or physiologically pulsatile conditions. This keeps the scope of the current manuscript focused on ultrasound-based SPION monitoring while acknowledging the relevance of X-ray PIV as a future complementary modality.

Comment 6:

The highest concentration (5 mg Fe/mL) is outside the therapeutic window. Clinical MDT doses are typically ≤0.5 mg Fe/kg, requiring supplementation with physiologically relevant concentrations (0.01–0.1 mg/mL). The impact of high concentrations on the cavitation signal saturation and PCM quantitative accuracy should be discussed.

Reply 6:

We thank the reviewer for this insightful comment. We agree that a concentration of 5 mg Fe/mL exceeds typical in vivo therapeutic levels, which are generally in the range of 0.01–0.1 mg Fe/mL before magnetic accumulation. In our phantom experiments, however, the reported concentrations represent local post-accumulation values at the tumor site, which can be significantly higher than systemic doses during magnetic drug targeting. The upper concentration (5 mg Fe/mL) was therefore included as an experimental reference to explore the dynamic range and potential saturation behavior of the PCM and MMUS signal.

We have clarified this point in the revised manuscript and added a note in the Discussion indicating that at high particle densities the cavitation signal exhibits sublinear growth due to bubble shielding and attenuation effects, which can limit PCM’s quantitative accuracy. Physiologically relevant concentrations (0.01–0.1 mg Fe/mL) remain within the linear response range and are therefore most suitable for clinical translation.

Comment 7:

Fusion consistency has not been verified. A validation target should be set at the tumour-vascular interface to compare the localization error between single-modality and fusion imaging (Bland-Altman analysis is recommended).

Reply 7:

We thank the reviewer for raising the issue of fusion consistency. In this study, the primary goal was to demonstrate that MMUS and PCM provide complementary visibility of tissue-bound versus circulating SPIONs within the same anatomical model, which is a prerequisite step before quantitative fusion-error validation can be meaningfully performed. A comparative accuracy study (e.g., Bland–Altman or landmark-based interface error estimation) would require a dedicated fusion/registration pipeline and fiducial targets engineered into the phantom specifically for co-localization benchmarking, which was beyond the scope of the present work.

However, we agree that such validation represents the logical next step toward translation. We have therefore added a statement to the Discussion/Future Work clarifying that future work will introduce ground-truth fiducial markers at the tumor–vessel interface to enable quantitative assessment of fusion accuracy and potential biases between MMUS and PCM, including Bland–Altman-style agreement analysis.

Comment 8:

Insufficient control experiments led to weaker evidence. Key control groups were missing, such as:

Reply 8:

â–¶ Non-magnetic nanoparticles (e.g., liposomes): to verify the specificity of MMUS;

We appreciate this suggestion. We agree that in principle a non-magnetic nanoparticle control could be used to demonstrate the specificity of MMUS; however, this aspect has already been comprehensively established in the MMUS literature. The displacement signal fundamentally arises from the magnetically induced force term, and the absence of susceptibility (χ ≈ 0) in non-magnetic particles has repeatedly been shown to result in no measurable magnetomotive response. To reflect this more clearly, we have added supporting citations in the revised manuscript to prior studies demonstrating that: (i) purely acoustic or flow-induced displacements do not generate a coherent phase-locked magnetomotive signal, and (ii) only magnetically susceptible particles give rise to frequency- and phase-matched displacement patterns.

Because the present manuscript does not propose a new contrast mechanism, but a new dual-modality integration strategy, we believe repeating this well-established control would not change the interpretation of the results. Nevertheless, to address the reviewer’s concern, we now explicitly state in the Discussion that future in vivo validation stages would incorporate non-magnetic particle controls as part of a broader specificity/sensitivity assessment.

â–¶ Microbubble contrast agents: to rule out the possibility that the PCM signal is due to residual gas;

We thank the reviewer for this comment. The concern that PCM signals could stem from residual gas rather than from the nanoparticles themselves is valid in general. However, in our previous work we have already performed reference comparisons using both positive cavitation controls (e.g., talcum microparticles) and negative controls (ultrapure degassed water), demonstrating that LA-SPIONs reliably produce stable and inertial cavitation under the same exposure conditions, while degassed media do not. These results confirm that the observed cavitation originates from the nanoparticles rather than from incidental microbubbles or gas inclusions.

Since the objective of the present study is not to re-establish SPION cavitation physics but to integrate PCM with MMUS into a dual-modality imaging framework, we have not repeated the microbubble control experiments here. To clarify this for the reader, we have added a sentence in the Discussion explicitly referring to the prior validation of SPION-driven cavitation and noting that clinically oriented future studies will include contrast-agent baselines as part of a broader safety/parameter-control evaluation.

â–¶ Ex vivo tissue validation: to demonstrate that the phantom conclusions can be extrapolated to biological tissue.

We appreciate the reviewer’s suggestion. The goal of the present study was to isolate and evaluate the dual-modality imaging performance (MMUS + PCM) under controlled and repeatable conditions rather than to characterize tissue biomechanics per se. For this reason, we employed PVA cryogel phantoms, which are widely used as validated tissue surrogates in both MMUS and cavitation imaging studies due to their reproducible elastic modulus, acoustic coupling properties, and stable long-term water content. These phantoms represent a standard intermediate step before ex vivo or in vivo studies and are routinely accepted in the field as a mechanistically appropriate approximation of soft tissue for magnetomotive displacement and acoustic cavitation.

Furthermore, the applicability of MMUS and PCM to biological tissue has already been demonstrated in earlier literature, which we now cite explicitly in the manuscript. Since the contribution of this work lies in combining the two modalities into a unified framework for complementary tissue-bound vs. circulating SPION detection, rather than repeating prior demonstrations in excised tissue, we consider the phantom-based validation sufficient for the scope of this study. We have also added a note in the Discussion section clarifying that the next step toward translation will involve ex vivo evaluation followed by in vivo studies.

Round 2

Reviewer 1 Report

Comments and Suggestions for Authors

I am satisfied by the authors' revisions and believe this manuscript is now acceptable for publication.

Author Response

Comment: I am satisfied by the authors' revisions and believe this manuscript is now acceptable for publication.

Reply: We sincerely thank the Reviewer for their positive feedback and are grateful for the acceptance of our manuscript.

Reviewer 3 Report

Comments and Suggestions for Authors

I have carefully checked the revised version of the manuscript, and I consider that the authors have made great efforts to improve the content of the manuscript and have well addressed the comments proposed by the reviewers. The current version of the manuscript can be accepted without further comments.

Author Response

Comment: I have carefully checked the revised version of the manuscript, and I consider that the authors have made great efforts to improve the content of the manuscript and have well addressed the comments proposed by the reviewers. The current version of the manuscript can be accepted without further comments.

Reply: Thank you for your thorough review and positive assessment of our revised manuscript. We appreciate your acknowledgement of the efforts made to improve the content and address the reviewers’ comments. We are grateful for your recommendation for acceptance.